# Analysis of Important Fabrication Factors That Determine the Sensitivity of MWCNT/Epoxy Composite Strain Sensors

**DOI:** 10.3390/ma12233875

**Published:** 2019-11-24

**Authors:** Mun-Young Hwang, Lae-Hyong Kang

**Affiliations:** 1Department of Mechatronics Engineering, Jeonbuk National University, 567 Baekje-daero, Deokjin-gu, Jeonju-si 54896, Korea; munyoung.h@jbnu.ac.kr; 2LANL-JBNU Engineering Institute-Korea, Jeonbuk National University 567 Baekje-daero, Deokjin-gu, Jeonju-si 54896, Korea; 3Department of Flexible and Printable Electronics, Jeonbuk National University 567 Baekje-daero, Deokjin-gu, Jeonju-si 54896, Korea

**Keywords:** multi-walled carbon nanotubes, polymer–matrix composites, strain sensor, piezoresistivity

## Abstract

Composite sensors based on carbon nanotubes have been leading to significant research providing interesting aspects for realizing cost-effective and sensitive piezoresistive strain sensors. Here, we report a wide range of piezoresistive performance investigations by modifying fabrication factors such as multi-wall carbon nanotubes (MWCNT) concentration and sensor dimensions for MWCNT/epoxy composites. The resistance change measurement analyzed the influence of the fabrication factors on the changes in the gauge factor. The dispersion quality of MWCNTs in the epoxy polymer matrix was investigated by scanning electron microscopy (SEM) images and conductivity measurement results. A configuration circuit was designed to use the composite sensor effectively. It has been shown that, in comparison with commercially available strain gauges, composites with CNT fillers have the potential to attain structural health monitoring capabilities by utilizing the variation of electrical conductivity and its relation to strain or damage within the composite. Based on the characteristics of the MWCNT, we predicted the range of conductivity that can be seen in the fabricated composite. The sensor may require a large surface area and a thin thickness as fabrication factors at minimum filler concentration capable of exhibiting a tunneling effect, in order to fabricate a sensor with high sensitivity. The proposed composite sensors will be suitable in various potential strain sensor applications, including structural health monitoring.

## 1. Introduction

Since application studies on Carbon Nano-Tubes (CNT) have begun, CNT-based strain sensors are ideal measuring instruments for structural health monitoring (SHM) due to the superior mechanical properties of CNTs, i.e., elastic modulus (up to 1 TPa) and strength (up to 63 GPa), as well as their considerable conductivity and high aspect ratio (up to 1000). Because the CNT strain sensor has a composite type, it can be an ideal substitute for applications where metal strain gauges are difficult to apply. For the proper substitution effects by CNT-based composite sensors, the research cases of CNT/polymer sensors have a higher gauge factor than the conventional commercial metal foil strain gauge (gauge factor = 2). For composite sensors with CNT, the gauge factor should be defined according to the specified strain level, because they exhibit a nonlinear piezoresistive response [1,2,3,4,5].

The dispersion state of CNTs in the polymer matrix firmly decides the mechanical, electrical, and thermal properties of composites [6,7,8]. Notably, the electrical conductivity of composites filled with CNTs at the same CNT concentration can have two times differences of magnitude among the samples prepared using different dispersion methods [9]. The most commonly chosen methods to disperse CNTs in epoxy resin are ultrasonication or shear mixing [8,10,11]. The presence of robust Van der Waals forces and much aggregation among CNTs are some of the hurdles that render the effective dispersing of CNTs in a polymer matrix a lasting research issue [8,11]. Chemical treatments of CNTs, such as the usage of chemical surfactants and dilute solutions, have been found to make the dispersing process effective but did not necessarily have a positive effect on the electrical conductivity of the fabricated composites [12,13,14]. The shear mixing conditions during the mixture of the CNT/resin for composite production were observed to be effective on composite electrical conductivities and, in some cases, increased the electrical percolation threshold [15].

The dispersion state of CNTs in the epoxy matrix can be changed according to the preparation of the curing process and the condition of the curing process [16,17]. CNT morphologies and quantity of residual catalyst material can vary with synthesizing methods. The alignment of CNTs can yield improved mechanical and physical properties of composites in the direction of CNT orientation [18,19]. The formation of electrically conductive pathways within the composite is the main factor to determine the electrical conductivity due to the presence of CNT. CNT agglomerates are dragged along with the polymer matrix and pulled apart during the stretching of the composite. This behavior of CNT agglomerates occurs because the bonds between CNT agglomerates and the polymer matrix are stronger than the bonds within the CNT agglomerate [20]. If there is no failure such as permanent deformation or internal cracking, unloading in the composites repairs contact between the CNT agglomerates. Because the electrical resistivity of the composite is affected by the distance between the conductive particles, it can be used as a measure of strain, and indirectly, as a structural health monitoring tool.

Also, Wichmann et al. [21] investigated the sensitivity of CNTs/epoxy nanocomposites according to the direction of the bending deformations. It was proved that CNTs/epoxy nanocomposites could exhibit linear piezoresistive response with high reproducibility. Hu et al. [22,23] investigated the piezoresistivity effect on fabrication parameters such as the curing time and mixing speed by using the various numerical simulations and experimental measurements. It was shown that the sensor sensitivity increased in the condition of the lower curing temperature and higher mixing speed. Yin et al. [24] studied the dynamic strain response to Multi-Wall Carbon Nano-Tubes (MWCNTs)/epoxy nanocomposites with different aspect ratios and dispersion conditions. MWCNTs with short aspect ratios and curved shapes exhibit linear piezoresistivity responses, while MWCNTs with high aspect ratios and relatively straight shapes have been found to result in non-linear responses due to the dominant role of tunneling resistors. In another study [25], it was observed that, at low MWCNT concentrations, a non-linear piezoresistivity is observed due to the dominant effect of tunneling, while sensors at higher concentrations show enhanced linearity. The main factors affecting the piezoresistivity of the CNT/polymer nanocomposites include the type of polymer, i.e., the thermosetting material, the thermoplastic material, as well as the polymer viscosity, which determines the characteristics and manufacturing process of the nanocomposite [8]. Among the various polymers available as resins for composites [26,27,28,29,30,31], epoxy resins are chosen because of their excellent adhesion to many substrates and their high thermal and mechanical properties [32,33,34]. However, it is necessary to study the efficient manufacturing process to realize the MWCNT/epoxy nanocomposite-based strain sensor as the dispersion process is complicated because of the low solubility of MWCNT in epoxy resin and weak dispersion.

In summary, the critical factors that determine the sensitivity of MWCNT/epoxy sensors are concentration, type, length of MWCNT fillers, the dispersion state of the fillers, and the manufacturing process of the composite. Although several researches [15,35] studied the sensitivity of the electrical and mechanical properties of CNT/epoxy, a reliable model that could prove the effect of fabrication parameters and their interactions on the properties of composites is still needed, and the investigation of the effect of the dispersion method on the piezoresistive characteristics of the fabricated composite has also been desired. This study aims to analyze fabrication factors that can improve the performance of MWCNT/epoxy sensors. The analysis was made on the manufacturing process factors related to the direct mixing method, on the sensor fabrication factors, and on the type of filler, in order to achieve the goal in this study. In other words, through the results of this study, the information and the fabrication methods, which are possible to make the design to be suitable for the environments and the purpose for the application of the composite sensors, would be presented to the designer of the strain composite sensors.

## 2. Fabrication Process of MWCNT/Epoxy Composite Sensor with Basic Type

### 2.1. MWCNT Filler Dispersion Method

Considering only the dispersion of MWCNTs, an optimized process that can prevent the aggregation properly while maintaining the aspect ratio of MWCNTs should be studied. In this study, MWCNTs were dispersed via the physical dispersion method. The advantage of the solvent-based dispersion method is that, unlike the chemical dispersion method, the inherent conduction properties of CNTs are maintained. There are various methods to maintain uniform dispersion while maintaining inherent conductivity.

Shear-mixing [36,37,38] or ultrasonication [39,40,41,42] are usually selected for the dispersion of nanotubes in a solution state. The transfer of local shear stress, which breaks down the aggregates, is the main factor in these dispersion methods. So, the shear energy delivered to the nanotube-cluster to exceed the binding energy of the system is required for the separation of nanotubes. Although the dispersion state obtained may only be a short one, it significantly supports the surface adsorption of interfacial particles such as surfactants and compatible solvent molecules, which may consequently stabilize the dispersion of CNTs. During processing, the level of energy density transferred into a solution is equivalent to the shear stress achievable. 

Mechanical shear-mixing can be performed in both low-viscosity solvents, or highly viscous polymer melts. The fluid strain rate for the conventional melt shear mixing is reliant on the rotational velocity of the mixer blade. Note that if one wishes to shear-mix in a low-viscosity solvent, the shear stress delivered to the CNT aggregates can drop, offering very little expectation of achieving dispersion [43,44]. This view is that shear mixing is only suitable for dispersion of MWNT clusters in high-viscosity polymer melts. Compared to mechanical shear mixing, ultrasonication uses a very different mechanism to give the shear stress for dispersing aggregates. Cavitation occurs in a low-viscosity fluid above a certain ultrasonic intensity in the low-pressure regions of the transported wave. Once created, the cavitation bubbles collapse, causing an extremely high strain rate in the fluid in the proximate regions of bubble implosion [45,46]. The distribution of cavities is controlled by the geometry of the sonicator and the sonication settings and is inhomogeneous throughout the solution [47]. Taking a typical low-viscosity solvent, the localized shear stress imparted in the region of an imploding bubble can approach high energy.

Therefore, following previous studies, we have investigated a process for the proper dispersion of MWCNTs dispersed by sonication in low-viscosity solvents on high-viscosity epoxy. That is, the dispersion methods without chemical treatment were compared to maintain the dispersion of pre-dispersed MWCNTs in the curing process. We investigated the optimum dispersion process in the dispersion method based on the solvent for improving the sensitivity of the MWCNT/epoxy composite sensor by comparing the ultrasonic treatment method and the method involving a high-speed spin mixer. Two dispersion methods were compared. In the first method, MWCNTs were dispersed in an acetone solvent for 1 h using only a horn type-ultrasonic processor (Q125-220, Qsonica, Newtown, Connecticut, USA) with 125 W. The second method involved the dispersion of MWCNTs in acetone solvent using a high-speed mixer (8000D Mixer Mill, SPEX, Metuchen, New Jersey, USA) with a mixing speed of 1725 Revolutions Per Minute (RPM). The fabricated specimens were classified based on the dispersion method and the MWCNT addition concentration (0.3, 0.5, 0.8, and 1.0 wt%). The piezoresistivity and conductivity of all the specimens were compared to investigate the relation between the concentration of the MWCNTs and the sensitivity of the sensors.

### 2.2. MWCNT/Epoxy Composite Fabrication

The MWCNT (CM-150, Hanwha Chemical, Seoul, Korea)/epoxy (KFR-120, KUKDO chemical, Seoul, Korea) composite sensor was fabricated, as shown in Figure 1. The MWCNT/epoxy mixture filled in the mold with 60 mm × 10 mm × 1.5 mm and was also cured for 8 h in the vacuum bag at 80 °C as shown in Figure 2a to minimize the pores which could interrupt complete curing and which could deteriorate the physical properties. It was challenging to insert the mixture homogeneously into the mold since the mixture with an additional amount of over 0.5 wt% MWCNT was very viscous. So, the fabrication process using the vacuum bag helped the homogeneous insertion of the mixture because the vacuum bag could make even pressure in the pattern. Before the test of the sensor characteristics, the surface of the specimens was mechanically polished to minimize the influence of surface flaws, mainly the pores. In addition, to investigate the effect of the sensor dimension, as shown in Figure 2b, 0.5 wt% MWCNTs sensors which had an area of 10 mm × 10 mm were made with different thicknesses of 0.5 mm, 1.5 mm, and 2.0 mm in the condition of the same fabrication process. The thickness of the screen controlled the thickness of each specimen. Three specimens in the same fabrication condition were fabricated and tested to ensure the reliability of the experimental results, as shown in Figure 3a. A sensor electrode was fabricated by using a silver paste (P-100, CANS, Tokyo, Japan) and conductive epoxy adhesive on both ends of the fabricated composite sensor, as shown in Figure 3b. It should be noted that the electrodes of the sensor are correctly connected to reduce the signal noise by the high operating voltage.

## 3. Assessment of the Importance of Fabrication Factors in MWCNT/Epoxy Composite Sensor

### 3.1. Experimental Method: Analysis of Filler Concentration, Sensor Dimensions, and Dispersion Method

Changes in MWCNT/epoxy sensor sensitivity to the concentration of the filler added in the sensor, the thickness of the sensor, and the dispersion method were observed. The sensitivity of the MWCNT/epoxy sensor was determined from the variation of the resistance of the sensor due to the constant deformation of the structure attached with the sensor, as shown in Figure 4. That is, the changed sensitivity by the variation of the resistance was evaluated after the sensor with the changed fabrication factor was applied to the acrylic cantilever at the same deformation amount.

The resistance variation was measured according to a constant strain by connecting a multi-meter (8848-A, FLUKE, Everett, Washington) to the fabricated electrode. The strain of the sensor was measured by deflecting the free end with an interval of 0.1 inch (2.54 mm) over the range of 1 inch (25.4 mm) by using a fixing device. A gauge factor was used to quantitatively evaluate the sensitivity of the sensor as resistance changed. Gauge factors changed by the varied design parameters were compared in the compressive direction and tensile direction in the range of 2200 με.

The gauge factor is defined as the quotient of the relative resistance change ∆*R*/*R*_0_ and the applied strain as Equation (1):(1)$$\mathrm{Gauge}\;\mathrm{factor}=\frac{\Delta R/R_{0}}{\varepsilon }$$
where ∆*R* is the change in resistance and *R*_0_ is the initial resistance, which is used to evaluate the relative resistance change of the MWCNT/epoxy sensor. To measure the strain of the sensor owing to the deflection of the free end, we used Equation (2) based on cantilever theory [48] to calculate the strain at the location where the sensor is attached.

The strain calculated using Equation (2) is 0.002108 με at the location where the MWCNT/epoxy sensors and the commercial metal foil strain gauge were attached. The strain measured using the attached commercial strain gauge was 0.002014 by using the commercial strain indicator. The analysis strain result under the same condition in ANSYS software is 0.002125 με in the same measuring position. Because the error occurs after the third significant digit in all the three methods, it can be possible to apply Equation (2) to the strain measurement of MWCNT/epoxy composite sensor and to determine the gauge factor of the sensor. Experimental comparison of the gauge factor was done using Equation (2) and the resistance change was measured according to the manufacturing conditions of the specimens prepared in Section 2 above. The first compares the sensitivity with concentration at the same sensor area; the other compares sensitivity with varying sensor dimensions at the same concentration.
(2)$$\epsilon =\frac{\delta }{E}=3\times c\left(L-x\right)\times \;\frac{\delta }{L^{3}},$$
where *δ* is the deformation; *E* is the Young’s modulus; *c* (= *t*/2) represents the distance from the center axis to the top in the section, where *t* represents the thickness of the beam; *L* is the length of the cantilever; *x* is the distance from the fixed end to the center of the measured sensor.

Finally, we compared the sensitivity of commercial strain gauges with the MWCNT/epoxy specimen showing the best sensitivity results. In addition to sensitivity to static deformation characteristics, sensitivity to dynamic impact properties was also examined by the frequency response characteristics, as shown in the experimental setup of Figure 5. An amplifier circuit and a noise filter were designed to measure small resistance changes because the MWCNT/epoxy sensor cannot use a standard strain gauge indicator due to the high driving voltage. The designed circuit consisted of an AD620 amplifier, RG to determine the gain of the amplifier, and a bridge circuit consisting of one MWCNT/epoxy sensor and three fixed dummy resistors. Variable resistors were also used to fine-tune the dummy resistors inside the Wheatstone bridge. Commercial strain gauges connected to a commercial indicator and a laser Doppler vibrometer (LDV) were installed at the same time for the reliability of the measured data in the constructed circuit.

### 3.2. Simulation Method: Analysis of Filler Concentration and Filler Characteristics

The electrical conductivity of MWNCT/epoxy composites is an essential factor in controlling sensor performance. The electrical conductivity of the composites is mostly determined by the dispersed state and concentration of the conductive filler.

The dispersion method used has a significant influence on the dispersion state. However, it is impossible to control the dispersion of conductive fillers completely experimentally. Therefore, this study presents a simulation method that can predict the range of conductivity of composites that can be determined by the concentration, length, and orientation of MWCNT fillers under the conditions of the fillers varied by the dispersion method. Some theories can accurately predict the conductivity using existing percolation techniques [49,50,51,52]. These methods concentrate on predicting the accurate electrical conductivity using information that is already known, such as shape and the concentration of the MWCNT filler. On the other hand, the method presented in this study focuses on predicting the range of conductivity of the composites that can be formed from the concentration, length, and conductivity of MWCNT fillers. In terms of the design of the sensor, it is the method that can help in selecting sensor fabrication factors by reducing the range of filler species that can satisfy the required conductivity of the composite. If the proposed method is utilized, the consumption time of the comparative experiment associated with which additive should be selected can be reduced before conducting accurate conductivity predictions and experiments.

In the simulation method of this study, the electrical paths were continuously generated from the randomly generated MWCNT on the bottom electrode of 3− to the surface of the top electrode according to the dispersion condition in a three-dimensional x-y-z Cartesian coordinate system as shown in Figure 6. The first random MWCNT segment was located on the x-y plane, and the subsequent MWCNT segment was generated randomly at the ending point of the first MWCNT segment in the thickness direction. When MWCNT segments were in contact with the opposite surface, the contact point was noted.

When a conductive path was formed completely, the new MWCNT segment of the subsequent conductive path was generated randomly on the bottom electrode. The simulation was terminated when the volume fraction of the MWCNT segments reached the ratio of the additional amount in a unit cell. MWCNTs were randomly dispersed in the unit cell of (Ux × Uy × Uz). MWCNTs were assumed to have a diameter *D* of 15 nm based on the reference in which the same MWCNTs (CM-150) were used [53] and length *L* in a range of 1–10 µm through random determination for considering the MWNCT segments cut arbitrarily by a dispersion method which can be selected by the manufacturer. All the MWCNTs were assumed to be line segments and the adjacent MWCNT segments can grow and rotate in accordance with the setting conditions: (1) 360° rotation condition, considering that CNTs can grow anywhere; (2) 180° rotation condition, considering that MWCNTs can grow only upward; (3) aligned condition, considering that MWCNTs can grow on the plane on which the first CNT is located.

Each MWCNT segment is described using the starting ($x_{0},\;y_{0},\;z_{0}$) and ending ($x_{1},\;y_{1},\;z_{1}$) points, which can be expressed as follows:(3)$$\begin{array}{c} x_{1}^{i}=x_{0}^{i}+l^{i}\left(sin\theta_{i}cos\emptyset_{i}\right),\\ y_{1}^{i}=y_{0}^{i}+l^{i}\left(sin\theta_{i}sin\emptyset_{i}\right),\\ z_{1}^{i}=z_{0}^{i}+l^{i}cos\theta_{i}, \end{array}$$
where *i* is the index of the *i*th MWCNT segment and l,∅, and θ are the length, and azimuthal and polar angles of the segments, respectively. It is assumed that the coordinates of the starting point and the orientation angles of the MWCNT follow Equation (4) as a uniform distribution.
(4)$$\begin{array}{c} x_{0}^{i}=U_{x}\times \mathrm{Rand},\\ y_{0}^{i}=U_{y}\times \mathrm{Rand},\\ z_{0}^{i}=U_{z}\times \mathrm{Rand},\\ \emptyset_{i}=2\pi \times \mathrm{Rand},\\ \theta_{i}=cos^{-1}\left(2\times Rand-1\right), \end{array}$$
where Rand is a random number in the range 0–1. The electrical conductivity was evaluated by identifying the connective network linking two opposite surfaces of the unit cell. After the resistor network is developed, the total resistance of the network $R_{MWCNT}$ can be obtained using Ohm’s law and Equation (5). In addition, the electrical conductivity $\sigma_{MWCNT/Epoxy}$ can be calculated using Equation (6). The average number of connections was confirmed during 100 simulations for each amount of MWCNT addition.
(5)$$R_{MWCNT}=\left(4L_{MWCNT}\right)/\left(\pi \sigma_{MWCNT}D_{MWCNT}^{2}\right),$$
(6)$$\sigma_{MWCNT/epoxy}=\left(1/R_{MWCNT/epoxy}\right)\left(U_{z}/S\right),$$
where $U_{z}$ is the height of the unit cell, $L_{MWCNT}\;$ is the total length of a conductive path, $D_{MWCNT}\;$ is the diameter of the MWCNT, $R_{MWCNT/Epoxy}\;$ is the total resistance of the composite sensor calculated by considering that all the MWCNT segments are connected as parallel resistors, and S is the area of the x-y plane. These numerical and iterative calculations were performed using MATLAB and can easily predict the range of conductivities that can be created by the added conductive fillers, even though calculations have been performed for a short time compared to conventional modeling of molecular behavior.

## 4. Results and Discussion

### 4.1. Sensor Fabrication Results According to Manufacturing Conditions in the Experimental Method

The results of sensor fabrication, according to fabrication conditions, were analyzed by various experimental methods. Morphological characteristics were observed through scanning electron microscopy (SEM) images, as shown in Figure 7, for comparing the dispersion state according to the dispersion method conducted in this study.

Two kinds of sensors fabricated by different dispersion methods at the same concentration of 1.0 wt% MWCNT were observed. Figure 7a is an image in the fracture plane of the specimen fabricated via ultrasonic treatment, and Figure 7b is an image in the fracture plane of the specimen fabricated by using a high-speed mixer. It was challenging to find an image as in Figure 7a with an even distribution of MWCNTs in the area observed in the specimen fabricated via ultrasonic treatment. It was easy to find MWCNTs that were not cut. In contrast, images like Figure 7b were easily found in the specimen fabricated by using a high-speed mixer. This phenomenon can be one of the proofs that even distribution is possible by the high-speed mixer method. However, since the SEM image shows only a very narrow region, it is difficult to make a complete analysis using only these results. Conductivities of MWCNT/epoxy were measured with all the concentrations of MWCNTs according to the dispersion method in order to confirm the relationship between these morphological characteristics and actual sensor characteristics. Since the conductivity of the composite can vary widely depending on various experimental conditions, conductivities of MWCNT/epoxy composite via fabrication process similar to that used in this study are referred to [35]. Conductivity was calculated from the resistivity measured using a high resistance meter (AT-683, APPLENT, Jiangsu, China). 

Figure 8 shows that the specimens fabricated via the high-speed mixing method showed higher electrical conductivity than that of the specimens fabricated via the sonication method at all the concentrations of MWCNTs. When MWCNTs were only dispersed via sonication, they were not sufficiently dispersed. This result is the same as the morphological results. As the view of the [54] with a focus on the mixing conditions, high-speed shear mixing in a high-viscosity polymer melt can deliver an energy density. However, this energy level is still orders of magnitude lower than an ultrasonic cavitation event. For MWCNTs, one finds that high-viscosity shear mixing is able to separate the aggregates apart. Shear mixing is a better dispersion method because it can effectively separate the MWCNTs without causing damage to the filament. For an aspect ratio of 1000, which is relevant to the as-produced MWCNT sources, one notices that the fracture resistance of both SWNTs and MWCNTs is lower than the stress input level delivered by sonication. On the other hand, shear mixing is not able to achieve a stress level matching the binding energy density. Therefore, it can be said that the dispersion process selected in this study can maintain dispersibility during the curing process only for MWCNTs having a relatively long aspect ratio. These results are in agreement with the results of morphology and conductivity measurements.

In order to analyze the influence of the sensor performance on the manufacturing factors such as the concentration of MWCNTs, the dispersion method, and the sensor area, the gauge factors were compared according to the manufacturing factors by the strain measurement experiment described in Section 3.1. As shown in Figure 9 and Table 1, the results show that the gauge factors of the MWCNT/epoxy sensors with MWCNT concentration less than 0.5 wt% are higher than the gauge factor of the commercial strain gauge used in this study. For MWCNT weight fractions below 0.5 wt%, the average gauge factor increases with increased MWCNT content, indicating increased sensitivity as the MWCNT content increases. However, the gauge factors of the sensors above 0.5 wt% MWCNT are practically decreased, which may be due to the fact that the percolation network is already densely packed after 0.5 wt%. Furthermore, above 0.5 wt% MWCNT concentration CNT agglomeration within the host polymer may make further enhancement of the gauge factor difficult. Also, in the case of an intensive conductive network with high MWCNT loading, if a conductive path is broken down, the total resistance of the sensor shows a minor variation. However, for a sparse conductive network with a very low MWCNT loading, for a particular case of only two conductive paths, $\Delta R/R_{0}$ is at least approximately 50%, resulting in a higher gauge factor. Therefore, from the above results, because the piezoresistive characteristic is mainly explained by the tunneling effect by which electrons can be transferred between adjacent CNTs, a MWCNT/epoxy sensor with higher sensitivity can be fabricated at the minimum concentration in which the tunneling effect is exhibited. These results are the same as those of other studies [55,56].

The performance according to the thickness of the sensor was higher as the thickness was thinner, as shown in Figure 9c. The reason for this phenomenon is that the thinner the resin layer thickness gets, the more efficient the stress transmission between MWCNTs and matrix and by Poisson’s ratio becomes. The gauge factor presented in this study does not necessarily show higher values than the commercial strain gauge. The main reason is the use of thicker sensors. In the study of Sanli et al. [57], the sensor thickness was 118 µm. In contrast, the MWCNT/epoxy sensor used in this study has the smallest thickness of 500 µm. The thickness of the sensor used in the study of Tanabi et al. [58] for a similar process was 5000 µm. The most significant gauge factor presented in the study is 2.9. The reason for this phenomenon is that the polymer dominates the stress transfer in the composite sensor. However, the results of this study do not merely indicate that a thin sensor has high sensitivity. Despite its thicker thickness, it showed high sensitivity in the form of a rectangle. In other words, it was judged that the aspect ratio in the form of the sensor is applied as a vital factor of sensor sensitivity. In other words, the primary purpose of this study is to analyze the manufacturing factors affecting the sensitivity of the MWCNT/epoxy sensor. The performance of the sensor, according to the area of the sensor, was found to be higher as the area was broader, as shown in Figure 9d.

The reason for this phenomenon is that the sensor operating mechanism is a change in the conductive path and the resistivity by the tunneling effect. As the area of the sensor becomes extensive, the amount of the conductive path changed becomes more extensive, and the sensor sensitivity is improved by the increase of the resistance change amount. Figure 10 shows the relationship between the influences of the manufacturing factors mentioned so far. The most dramatic effect on the variation of the gauge factor is the concentration of the filler. In order to fabricate a sensor with high sensitivity from the relationship diagram, a sensor may require a large surface area and a thin thickness as fabrication factors at a minimum filler concentration capable of exhibiting a tunneling effect.

Furthermore, it can be confirmed that the gauge factors in the tension direction are higher than those in the compression direction. In the case of the 0.5 wt% MWCNT/epoxy sensor fabricated via ultrasonic treatment, the gauge factor in the tensile direction is 3.77 and that in the compressive direction is 3.14. For the reason of this phenomenon, when a compressive force is applied to a sensor, the decrease in the distance between CNTs is limited because neighboring CNTs cannot penetrate each other. In contrast, when a tensile force is applied, the increase in the distance between adjacent CNTs is not limited. In other words, under compressive strain, the piezoresistivity of MWCNT/epoxy saturates with the increase of applied strain; the sensitivity for the compressive strain is lower than tensile strain. Similar behavior was also observed elsewhere [22,23,25]. The effect of uneven dispersion can be confirmed in the results of Figure 9a where the specimen fabricated via sonication shows a broader error distribution than that of the specimens fabricated using the high-speed mixer in Figure 9b at all MWCNT addition concentrations. For example, in the ultrasonic dispersion method, the measured gauge factor range is extensive, even though the sensors are manufactured in the same batch (one mixture). However, in the high-speed mixer method, the measured gauge factors of the specimens fabricated in the same batch (one mixture) are relatively narrow. Although the resistance change of the specimen manufactured via sonication is large, the use of the high-speed mixing method would be preferred if the uniformity and repeatability of the manufacturing process were considered. Because the MWCNT/epoxy specimen showing the best sensitivity results was the 0.5 wt% MWCNT/epoxy composite sensor with 60 mm × 10 mm × 1.5 mm fabricated by the high-speed mixer method, the sensor was used in comparison with commercial strain gauges.

In the case of a sensor manufactured with a concentration of 0.5 wt% or less, because high current and voltage were required for driving the sensor owing to its higher resistance, it was not suitable for use in the configuration of the design circuit described in Section 3.1.

Figure 11 and Table 2 show the response characteristics of the sensor through free vibration using the impact hammer. From these results, the primary, secondary, and tertiary frequencies were confirmed at the same frequency in the response of the LDV, although there was a difference in their decibel values. Since the MWCNT/epoxy sensor is long and composite with dispersed filler, a relatively unclear signal was found compared to the signal of LDV. Future studies have found a need for research on methods to achieve high-resolution results in high-frequency response. However, the frequency response results of the MWCNT/epoxy sensor are similar to those of commercial metal foil strain gauges. From such a result, the verification of the operating circuit to utilize the sensor could be completed. It also proved that it could perform the same role as the strain gauge.

### 4.2. Analysis of the Range of Conductivity with the Selected MWCNTs in the Simulation Method

Using the resistance and diameter information of the MWCNTs used in this study, we predicted the range of conductivity that composites with MWCNT fillers could exhibit. In this study, it is assumed that the length and orientation of MWCNT are different according to the dispersion method considering only the physical dispersion. Figure 12 shows the correlation between the maximum length and the growth direction of MWCNTs grown with constant addition concentration. Each sheet has a constant value of concentration. The larger the maximum growth length of the MWCNTs, the greater the expected conductivity is, since the probability of generation of a conductive path in the unit cell increases. The more isotropic the MWCNTs in the unit cell are, the higher the predicted conductivity is because the probability of generation of a conductive path in the unit cell also increases. In Figure 12b, the effect of the maximum growth length of the MWCNTs is higher than that of the growth direction. To verify the simulated conductivity of MWCNT/epoxy sensor, we compared the measured conductivity of the sensor with the results of simulation performed 100 times according to the maximum growth length and growth direction of MWCNTs with the addition concentrations of 0.3, 0.5, 0.8, and 1.0 wt%. From the simulation results in Figure 13, it can be observed that the conductivity is relatively small when the maximum growth length is 1 µm. This result is consistent with the experimental results, which indicated that the length of MWCNTs is shortened owing to a very high shear force and very long mixing time, and it is difficult for the shortened MWCNTs to form a network [49,59].

In addition, the results of conductivity in the 360° range of growth direction of MWCNTs are remarkably low in Figure 13, and consequently, it can be concluded that when the MWCNT aggregation in the matrix is high the conductivity of the sensor may decrease sharply. This is consistent with the experimental results which indicated that if the shear force and mixing time are significantly reduced, fillers cannot be adequately dispersed, and the conductivity can be reduced [60]. From these results, it is proved that the range of conductivity that the selected filler can exert after composite molding can be predicted by using the simulation method presented in this study. There is a wide variation in the conductivity of the composites produced because it is impossible to control the conductive filler dispersion within the composite material completely. Because this deviation of the conductivity can be predicted by this simulation method, to realize the required conductivity of the sensor, this simulation method can help to choose how many fillers to use and how high dispersion intensity should be according to the selected filler.

## 5. Conclusions

In this study, the influence of fabrication factors in the MWCNT/epoxy composite sensor was evaluated, as shown in Table 3. We investigated the fabrication factors that can achieve optimal performance by analyzing the relation between the performance of the sensor and the variables in experiments. Experimental methods were used to investigate the fabrication parameters of MWNCT concentration, sensor length, and dispersion method. A measurement experiment of the gauge factor was designed to quantitatively evaluate the sensor sensitivity by resistance change of the sensor. The concentration of MWCNT fillers showed the highest effect in the experimental results. The optimal design of the MWCNT/epoxy composite sensor was to minimize the thickness and to maximize the surface area at the minimum MWCNT concentration capable of exhibiting the tunneling effect. In addition, morphology evaluation by SEM, conductivity measurement, and deviation of resistance measurement results were used for investigating dispersion state in the composite. Finally, MWCNT/epoxy sensor was compared with the metal strain gauge in the frequency response experiment to verify the application of the composite sensor. As a result, it has been confirmed that if a design circuit composed of an amplification circuit is used, it could sufficiently act as a strain sensor in structural health monitoring like a commercial metal foil strain gauge.

The simulation method was used to analyze the concentration and characteristics of MWCNTs. Based on the properties of the MWCNTs added, we predicted the range of conductivity that the MWCNT-dispersed composite could exhibit. The proposed simulation method makes it easy to predict what MWCNT conditions should be used for the required conductivity in terms of design. As a result, the larger the maximum growth length of the MWCNTs is, the higher the expected conductivity is, since the probability of generation of a conductive path in the unit cell increases. The more isotropic the MWCNTs in the unit cell are, the higher the predicted conductivity is, because the probability of generation of a conductive path in the unit cell also increases.

## Figures and Tables

**Figure 1 materials-12-03875-f001:**
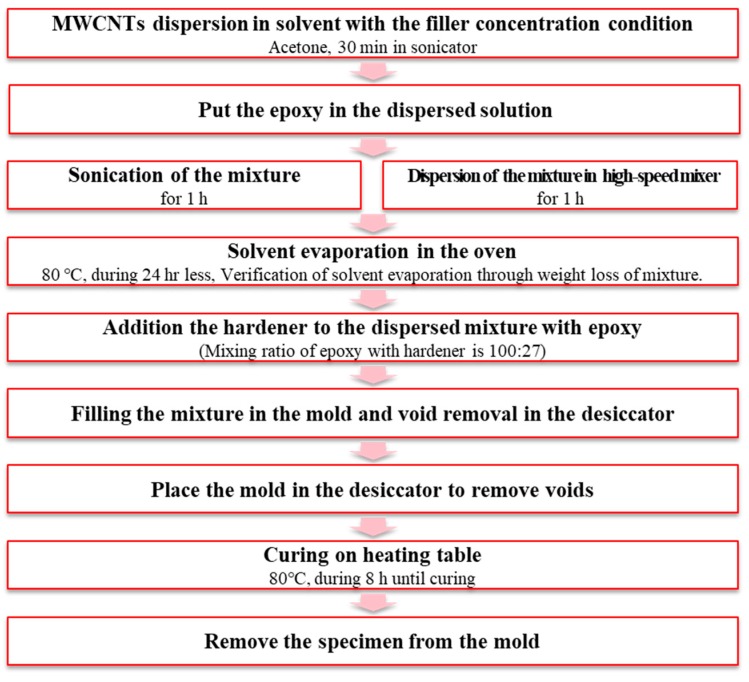
Fabrication procedure of the multi-wall carbon nanotube (MWCNT)/epoxy composite sensor.

**Figure 2 materials-12-03875-f002:**
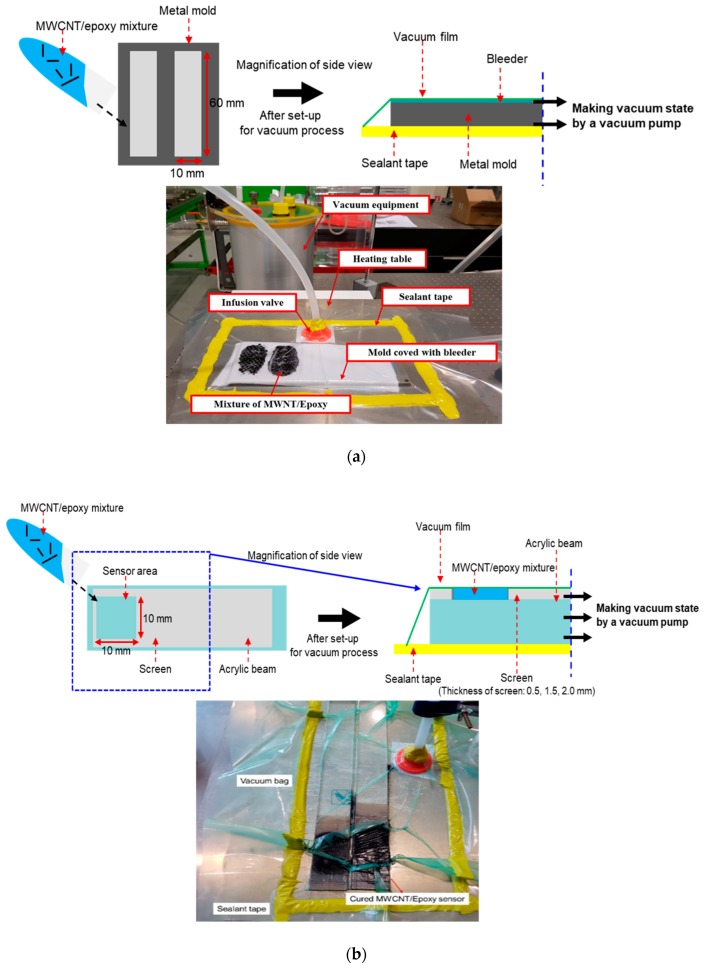
Fabrication of MCWNT/epoxy composite sensor (**a**) in mold and (**b**) on acrylic beam using vacuum-assisted molding.

**Figure 3 materials-12-03875-f003:**
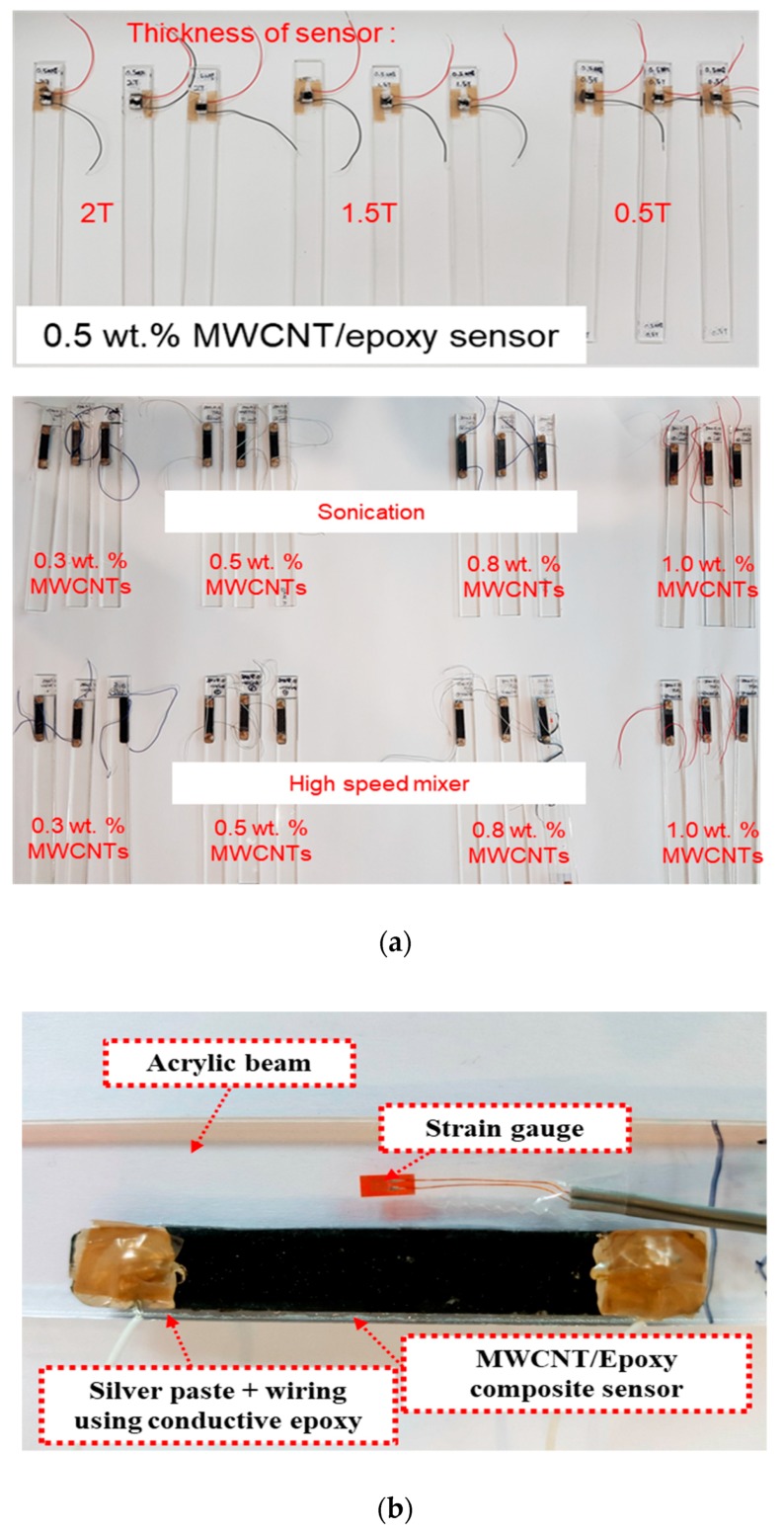
Specimens of the MWCNT/epoxy composite sensor with the thickness, (**a**) MWCNT concentration and dispersion method, and thickness; (**b**) the specimen attached on the acrylic beam for comparing with the commercial strain gauge.

**Figure 4 materials-12-03875-f004:**
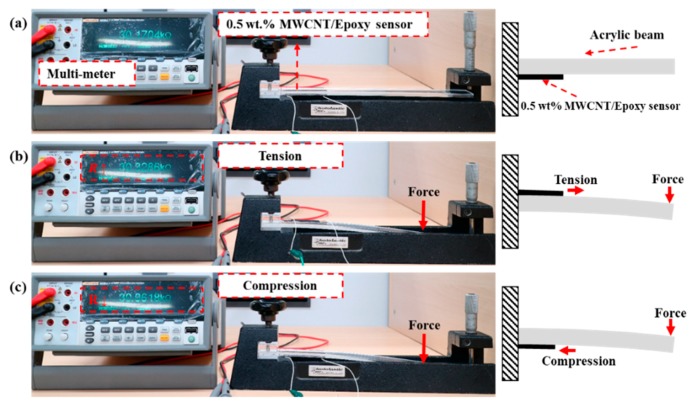
Resistance variation of MWCNT/epoxy composite sensor when deflection of the beam end occurs: (**a**) steady-state, (**b**) tension direction, and (**c**) compression direction.

**Figure 5 materials-12-03875-f005:**
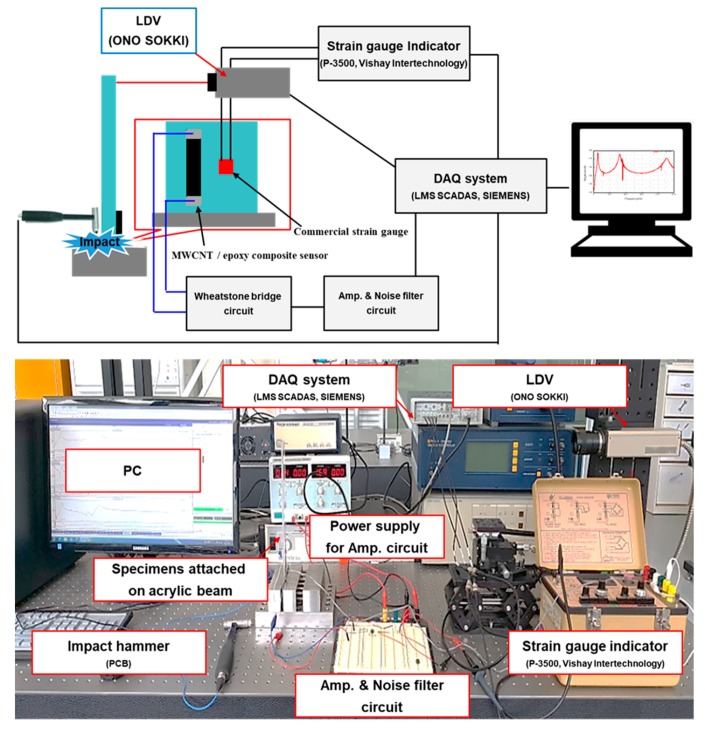
Schematic of the frequency response experimental setup of the MWCNT/epoxy composite sensor and the commercial strain gauge for verifying availability.

**Figure 6 materials-12-03875-f006:**
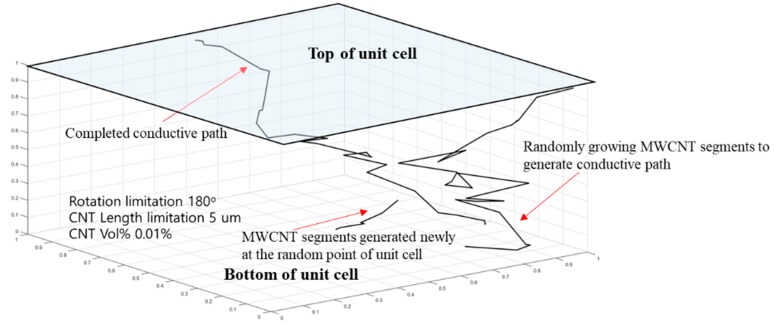
Schematic of a unit cell in simulation of predicting the range of electrical conductivity possible to be exhibited.

**Figure 7 materials-12-03875-f007:**
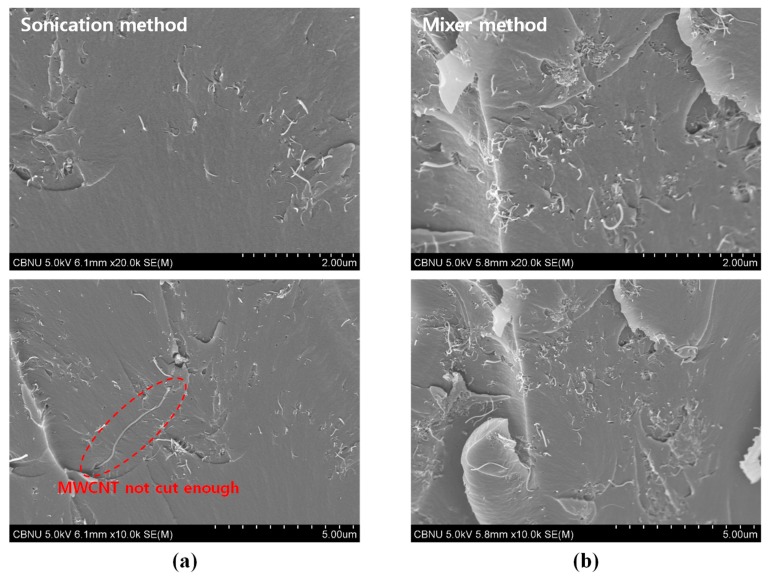
FE-SEM image of (**a**) 1.0 wt% MWCNT/epoxy sensor fabricated via the sonication method, (**b**) 1.0 wt% MWCNT/epoxy sensor fabricated via the high-speed mixer method.

**Figure 8 materials-12-03875-f008:**
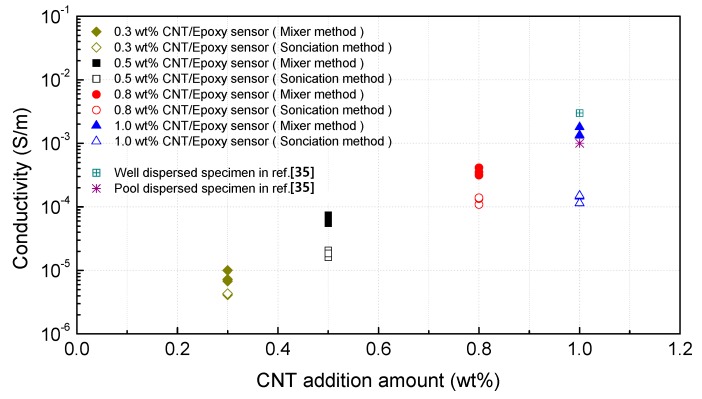
Measured electrical conductivity of various sensors according to the addition amount of MWCNTs and dispersion method.

**Figure 9 materials-12-03875-f009:**
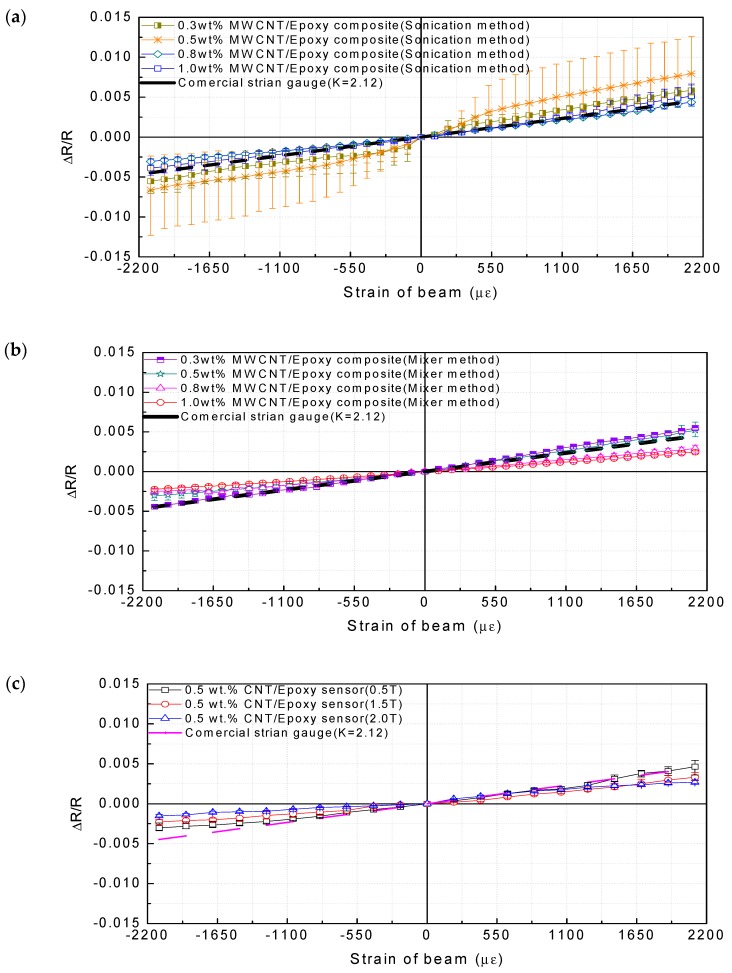
Results of the resistance changes in the sensor fabricated according to the tip deflection of the cantilever beam by (**a**) the sonication method; (**b**) the high-speed mixer method; (**c**) with changing thickness; and (**d**) with changing sensor area.

**Figure 10 materials-12-03875-f010:**
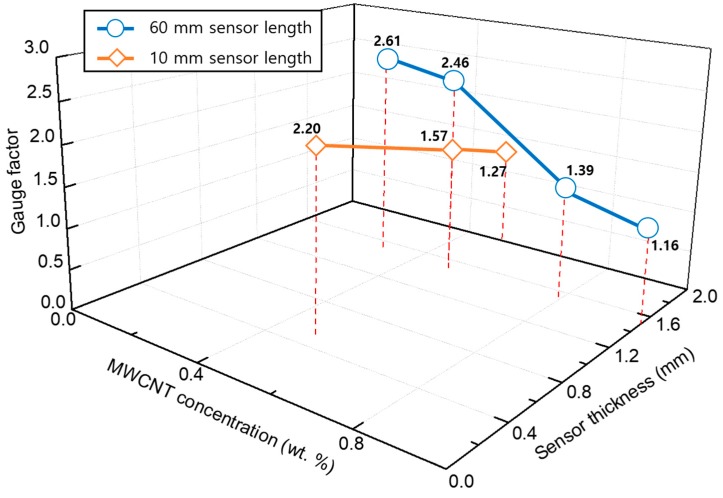
Gauge factor according to fabrication factors: the concentration of the MWCNT fillers, the thickness of the sensor, and the length of the sensor.

**Figure 11 materials-12-03875-f011:**
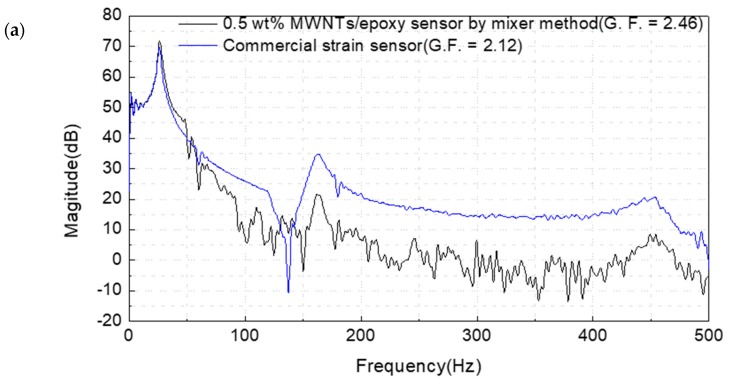
Comparison of frequency response results of acrylic beam (a) in 0.5 wt% MWCNT/epoxy sensor and strain gauge; (b) in LDV.

**Figure 12 materials-12-03875-f012:**
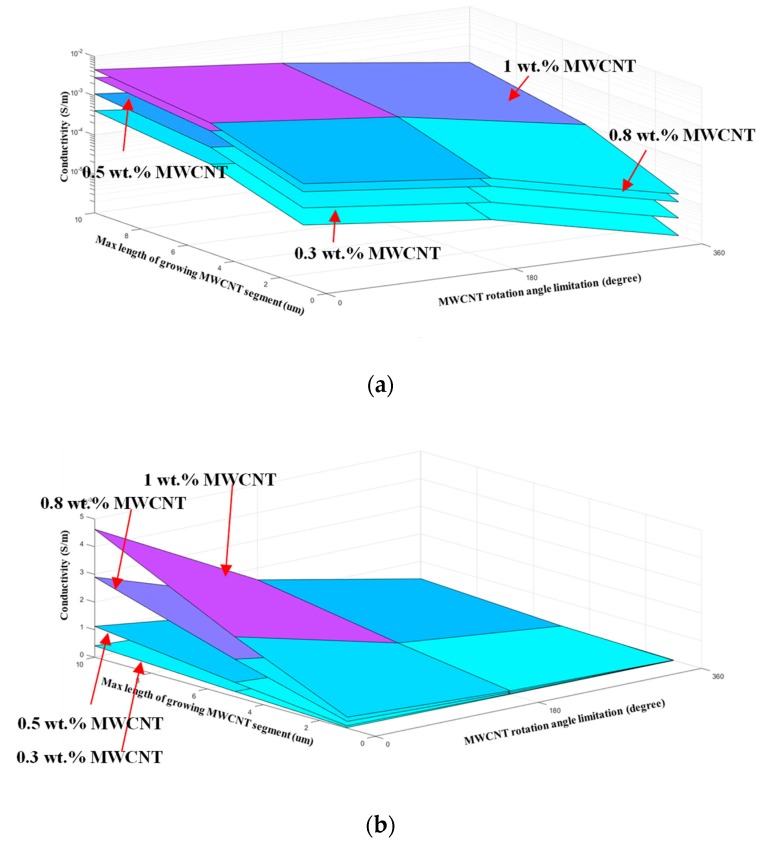
Conductivity according to the growth direction and maximum growth length of MWCNTs grown at constant addition concentration of MWCNTs; conductivity of MWCNT/epoxy sensor in (**a**) log scale, and (**b**) linear scale.

**Figure 13 materials-12-03875-f013:**
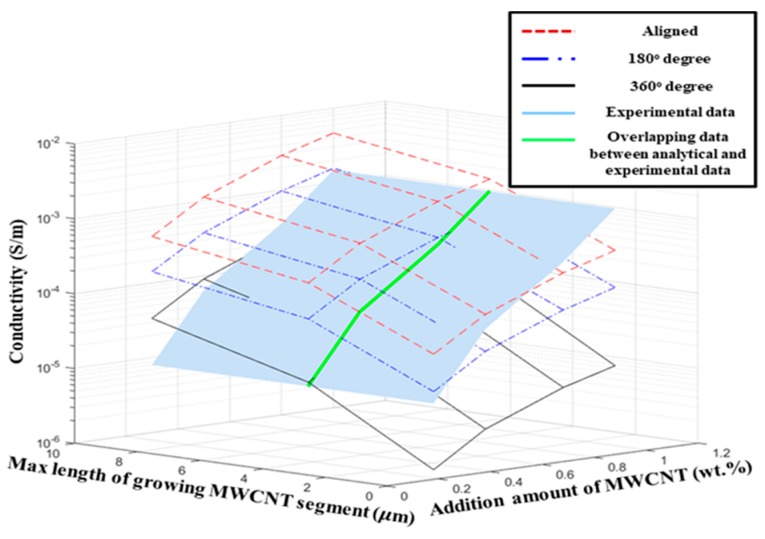
Comparison between the conductivity in the measured results and simulated results according to the maximum growth length of MWCNTs and the addition concentration of MWCNTs in the growth direction of MWCNTs.

**Table 1 materials-12-03875-t001:** Gauge factors according to fabrication factors which are concentration of the MWCNT fillers, the thickness of the sensor, and the length of the sensor.

MWCNT Concentration (wt%)	Gauge Factor in Compression Direction	Gauge Factor in Tension Direction	Dispersion Method	Sensor Dimensions (mm) (Horizontal × Vertical × Thickness)
0.3	−2.10	2.61	high-speed mixer	60 × 10 × 1.5
0.3	−2.62	2.78	sonication	60 × 10 × 1.5
0.5	−1.44	2.46	high-speed mixer	60 × 10 × 1.5
0.5	−3.14	3.77	sonication	60 × 10 × 1.5
0.8	−1.19	1.39	high-speed mixer	60 × 10 × 1.5
0.8	−1.45	2.07	sonication	60 × 10 × 1.5
1.0	−1.06	1.16	high-speed mixer	60 × 10 × 1.5
1.0	−1.84	2.44	sonication	60 × 10 × 1.5
0.5	−1.43	2.20	high-speed mixer	10 × 10 × 0.5
0.5	−1.08	1.57	high-speed mixer	10 × 10 × 1.5
0.5	−0.72	1.27	high-speed mixer	10 × 10 × 2.0

**Table 2 materials-12-03875-t002:** Comparison of frequency response results of acrylic beam in 0.5 wt% MWCNT/epoxy sensor, strain gauge, and laser Doppler vibrometer (LDV).

	MWCNT/Epoxy Sensor	Strain Gauge	Laser Doppler Velocimetry
1st Frequency	26.5 Hz	26.5 Hz	26 Hz
2nd Frequency	163 Hz	163 Hz	162 Hz
3rd Frequency	456.5 Hz	462.5 Hz	453.5 Hz

**Table 3 materials-12-03875-t003:** Recommended fabrication factors for high sensitivity sensor performance in MWCNT/epoxy composite sensor.

Fabrication Factors	Dispersion Method	MWCNTs Concentration	Sensor Design
Recommendation in this study	High-speed mixer (shear mixing)	Minimum concentration above the percolation threshold	Thinner, higher aspect ratio to thickness
Comparison in this study	With sonication	With four concentration types	With 4 design types
Comparison method	By experiments By simulation By reference [54]	By experiments By simulation By reference [55,56]	By experiments By reference [57,58]

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
