# Peer review of "Analysis of Important Fabrication Factors That Determine the Sensitivity of MWCNT/Epoxy Composite Strain Sensors"

_materials, 2019, doi:10.3390/ma12233875_

Round 1
Reviewer 1 Report
In this article, the author tries to clarify the influence of fabrication parameters on the strain sensing properties of the MWCNT/epoxy composite. Though the article is well-structured, it lacks systematic methodology and scientific explanations. The work needs a major revision before it is accepted for publication.
1. The author needs to make a deep literature study to understand the current state of the art in the relevant field. For example, in comparison with the below reference what is the innovation in this work?
Sanli, A. Benchirouf, Ch. Müller, O. Kanoun: “Piezoresistive performance characterization of strain sensitive multi-walled carbon nanotube-epoxy nanocomposites”, In: Sensors and Actuators A: Physical - Elsevier BV. - 254. 2017, pp. 61 – 68, 2017.
2. The author claims, in line 22 and in line 314 that the composite can serve as a strain sensor. Was this not expected?
3. Detailed information on the MWCNT in terms of length is missing. The author claims in line 197 the length assumed for simulation is 10 µm but in the SEM images in figure 7 the length is much smaller. Maybe a TEM image can help to support.
4. The author compares two approaches for dispersing CNTs in epoxy, but a detailed investigation is missing. Both approaches were done with the same duration and probably a fixed amplitude. However, they are different approaches with different level of energy transmitted into the medium. Detailed analysis of different duration and amplitude must be investigated to really understand the difference in these approaches. The amplitude of sonication is mixing.
5. Line 230, 0.1wt% was never mentioned, yet it appears here, figure 7 shows 1.0wt% -> which is correct?
6. In multiple instants, the author claims mixing is better than sonication but 1 hour of sonication could result in CNT breakage. This means the conductivity will be increased and fewer CNTs are visible in the SEM as there are smaller than in the other approach.
7. From table 1 it is evident that there is no significant change or improvement in the gauge factors across all methods. Even the conventional strain gauges have a gauge factor of around 2. Several works have been done in the same direction to improve the gauge factor and there is evidence that CNT based sensors exhibit high gauge factor-like 78 (in the above reference). What is the reason for this low gauge factor?
8. Line 291, the author explains the phenomenon as the results of the tunneling effect in CNTs, but degree of compression or tensile is a material (polymer) related factor. This needs to rephrased or sufficient literature must be provided to support the claim.
9. Line 286: Large surface area; how can two different shapes of different dimensions be compared? a rectangular geometry will have a different strain transferring mechanism compared to a square. To compare with the commercial strain gauge at least the sensor must be in the same thickness and not higher.
10. Line 286: thin thickness -> what is thin, the minimum investigated thickness is 500 µm which is not thin.
11. Sensor characterisation in terms of performance stability and life time is missing.
12. Figure 1, 2, 3a, 6, 10, 11, 12, 13 is not readable please improve
13. Line 102: sensing -> sensor
14. Line 259: addition amount -> concentration
Reviewer 2 Report
The objective of this paper is very interesting and the investigations are well carried out. The combination experimental works with simulation is very good. To improve qthe quality of the paper, I just suggest to emphasize the figures 2, 3, 4 and 5 because they are too small and not enough clear and explicit. Otherwise, the content is clear and very interesting.
Reviewer 3 Report
The Authors have presented a quite thorough work on investigating the applications of MWCNT in composites as strain sensor by
both experiment and simulation. It covers quite a few parameters that may help industry to make use of therefore it is
recommended to publish in Materials after addressing the following issues:
1. Line 233, it says that the composites made via high-speed mixer has poorer distribution than the sonication one, while line
251 opposite statement was made.
2. Several figures/charts are not clear, either too mall font or poor image quality, such as Figure 1/2/3/5/10/12/13.
3. It is well known that the sensitivity/reproducibility of CNT/composites sensor is one issue, just wonder if the authors have
any error bar in Figure 10, or any sort of confidence interval?
4. The authors claimed the sensor is comparable to commercial ones. It might be true if according to gauge factor only, but the
intrinsic conductivity is another issue, while the sensor made this work does not have a conductivity comparable to commercial
ones as stated in line 303.
5. It is good to present the simulated method and result as shown in the manuscript. However from the composite manufacturing
perspective, it might be easier to correlate with the spatial orientation, dimension and tangling of CNT. The authors are
suggested to elaborate on that.
6. 'Chapter 3' appears several times but it is a bit confusing, can use Section 3?
Reviewer 4 Report
The manuscript investigated the effects of fabrication factors on the sensitivity of MWCNT/epoxy composite strain sensors. Page 7 line 230, “concentration of 0.1 wt. % MWCNT” is incorrect. Why is the sensitivity of composite strain sensor with 0.5 wt.% of MWCNT better than those of 0.3 wt.%, 0.8 wt.% and 1.0 wt.% MWCNT/epoxy composite sensors. The sensitivities of compressive and tensile strains are different. How can the composite strain sensor be used to measure the free vibration subjected to both the compressive and tensile strains. Most of the sensitivity of the composite strain sensor is less than strain gauge except for 0.5 wt.% of MWCNT. Thus, the composite strain sensor presented in this work in not impressive in terms of strain sensitivity.Author Response
Please see the attachment

Round 2
Reviewer 1 Report
The article needs some minor formatting issues to be addressed:
Font type mismatch across figures and text Figure 9: x-axis could be represented in %Reviewer 4 Report
I am satisfied with the responses made by the authors. The manuscript can be published as it is.